# LEARNING MODAL-MIXED CHAIN-OF-THOUGHT REASONING WITH LATENT EMBEDDINGS

## ABSTRACT

We study how to extend chain-of-thought (CoT) beyond language to better handle multimodal reasoning. While CoT helps LLMs and VLMs articulate intermediate steps, its text-only form often fails on vision-intensive problems where key intermediate states are inherently visual. We introduce modal-mixed CoT, which interleaves textual tokens with compact visual "sketches" represented as latent embeddings. To bridge the modality gap without eroding the original knowledge and capability of the VLM, we use the VLM itself as an encoder and train the language backbone to reconstruct its own intermediate vision embeddings, to guarantee the semantic alignment of the visual latent space. We further attach a diffusion-based latent decoder, invoked by a special control token and conditioned on hidden states from the VLM. In this way, the diffusion head carries fine-grained perceptual details while the VLM specifies high-level intent, which cleanly disentangles roles and reduces the optimization pressure of the VLM. Training proceeds in two stages: supervised fine-tuning on traces that interleave text and latents with a joint next-token and latent-reconstruction objective, followed by reinforcement learning that teaches when to switch modalities and how to compose long reasoning chains. Extensive experiments across 11 diverse multimodal reasoning tasks, demonstrate that our method yields better performance than language-only and other CoT methods. Our code and data will be publicly released.

## 1 INTRODUCTION

Chain-of-thought (CoT) (Wei et al., 2022; Zhou et al., 2022) has enabled large language models (LLMs) and vision-language models (VLMs) to generate intermediate reasoning steps and substantially improve performance on complex tasks (Chen et al., 2023; Zhang et al., 2024). By encouraging models to articulate stepwise deductions, CoT can better elicit LLMs and VLMs to generate accurate and faithful responses. However, its language-only form struggles on multimodal problems (*e.g.,* 3D spatial reasoning and visually grounded logical queries (Zhu et al., 2024)), where crucial intermediate states are inherently *visual* rather than textual. Many multimodal tasks require mental rotation and transforms, or fine-grained spatial relations that are cumbersome to describe in words. These limitations become more acute under vision-intensive and long-horizon spatio-temporal dependencies, where purely verbal descriptions fail to capture evolving visual context (Gao et al., 2025).

Humans address these cases by engaging *mental imagery* (Pylyshyn, 2002; Richardson, 2013): we sketch, reposition, and manipulate latent visuals in mind, externalizing only the pieces that matter for multi-hop reasoning (Yang et al., 2025). Cognitive studies describe a visuospatial "sketchpad" that complements verbal working memory, enabling rapid simulation of object poses, trajectories, and constraints with minimal verbalization. This latent, modality-interleaved process lets us flexibly switch between words and pictures, preserving speed while maintaining grounding. Motivated by this paradigm, we seek to equip VLMs with *modal-mixed* CoT: the ability to interleave textual tokens with compact visual "sketches" represented as latent embeddings. Such embeddings act as lightweight, model-native carriers of visual state, allowing the reasoning process to offload visual details to condensed latent embeddings while reserving language for high-level logic control. In doing so, we aim to reason beyond what text can describe, improve grounding, and make complex vision-intensive reasoning tasks both more accurate and more efficient.

A central challenge is the modality gap: inserting a new visual generation pathway must not erode the VLM's original linguistic knowledge and capabilities (Yi et al., 2025). Therefore, we should guarantee that the visual latent space is both *aligned* with the VLM's internal representations and *predictable* from its outputs. First, inspired by recent VLM-as-encoder ideas (Yang et al., 2025; Wu et al., 2025), we use the visual encoder and connection layer from the VLM itself as the latent embedding encoder. Specifically, we train the VLM to reconstruct its visual encoder produced embeddings for intermediate images within the modal-mixed chain-of-thought. In this way, the generated latent visual "rationales" will be in the same semantic space as its native features. This alignment makes the latent embeddings easy for the VLM to generate and consume. Second, we attach a diffusion-based latent decoder that is triggered by a special *start* token and conditioned on the VLM's hidden states to produce continuous visual embeddings. Benefiting from the strong vision generation ability, the diffusion model is able to carry the burden of fine-grained perceptual details, while the VLM focuses on supplying compact, high-level intent. Such a disentanglement-based design simplifies the function of the VLM and eases its learning objective, reducing the risk of catastrophic forgetting caused by overfitting visual details.

Building on these components, we adopt a two-stage training strategy for modal-mixed CoT. We first perform supervised fine-tuning on curated traces that interleave text and latent visuals to teach the format and basic skills. The objective couples next-token prediction with latent reconstruction, so the model learns *what* to sketch and *how* to reference those sketches later. Then, we apply reinforcement learning on the VLM to learn *when* to switch modalities and *how* to compose textual and latent steps for complex problems. Extensive experiments across 11 multimodal tasks demonstrate consistent gains over language-only CoT and strong improvements on complex visual reasoning tasks.

We summarize the key contribution of this paper as:

• We devise an architecture that integrates VLM with a diffusion model based decoder, to support modal-mixed reasoning with latent visual embeddings.

• We leverage a two-stage training strategy that first learns the modal-interleaved reasoning paradigm via supervised fine-tuning, then reinforcement learning for further adapt the two modes.

• Extensive experiments on 11 multimodal tasks have shown the effectiveness of our approach.

## 2 RELATED WORK

**Vision-language Models.** VLMs are adapted to interpret visual information to understand multimodal documents, forms, and charts where text and images are intertwined. Modern VLMs typically adopt a three-component architecture consisting of a vision encoder, a foundational LLM, and a connector module. Pioneering models such as Flamingo (Alayrac et al., 2022), BLIP (Li et al., 2023), and LLaVA (Liu et al., 2023a) were instrumental in establishing this paradigm for tasks like image captioning and visual question answering. Recent works have focused on refining the training process. A surge of work scales up the training data of VLMs and observes the improved performance on the reasoning ability, *e.g.,* KOSMOS-2 (Peng et al., 2023), InternVL2.5 (Chen et al., 2024b), and Qwen2.5VL (Bai et al., 2025). In addition, advanced post-training techniques like visual instruction tuning (Liu et al., 2023b) and reinforcement learning (Chen et al., 2025c) have also been applied in VLM to enhance the task solving capabilities. However, since VLMs can only generate language-based rationales, they do not perform well on vision-intensive tasks, *e.g.,* spatial and multi-image reasoning (Chen et al., 2025a). Although recent works have proposed unified generative models that can produce either text or vision outputs, their benchmark performance is still not as better as state-of-the-art VLMs (Wang et al., 2025b). In this paper, we adapt a VLM into a multimodal reasoner that can perform interleaved reasoning in vision and text modals. We simplify the vision generation task into producing high-level visual latent embeddings.

**Chain-of-thought Reasoning.** Chain-of-thought (CoT) reasoning has been widely explored as a way to enhance LLM performance by explicitly generating intermediate reasoning traces before producing final predictions (Wei et al., 2022; Khot et al., 2022; Zhou et al., 2022; Yue et al., 2023; Yu et al., 2023; Wang et al., 2024; Havrilla et al., 2024). Recent advances show that large-scale RL training can encourage models to generate much longer CoT and achieve stronger performance through test-time scaling, as demonstrated by OpenAI O1 (Jaech et al., 2024) and DeepSeek R1 (Guo

et al., 2025). Beyond simple linear chains, researchers have extended CoT into tree-structured reasoning paradigms (Yao et al., 2023; Xie et al., 2023; Hao et al., 2024a). CoT methods are also widely used in VLMs for solving multimodal reasoning tasks (Zhang et al., 2024). Recent works have proposed to use interleaved CoT for solving vision-intensive tasks (Gao et al., 2025). By equipping with visual tools (*e.g.,* zooming in and drawing lines), VLMs can learn to interleave the tool use and text generation, to edit the input for composing intermediate image to guide reasoning (Wang et al., 2025a). However, such a way cannot handle open questions that might require special operations on the images (*e.g.,* drawing irregular masks). To solve it, we aim to internalize the image generation process into producing latent visual embeddings, which enable the VLM to learn generating high-level semantics and also reduce the inference latency from tool invocation.

**Latent Reasoning.** Another line of recent works investigates latent reasoning, where the reasoning process happens in hidden states rather than being explicitly generated in language. Methods such as introducing special latent tokens (Goyal et al., 2023; Pfau et al., 2024), knowledge distillation (Deng et al., 2023; 2024; Yu et al., 2024), or architectural modifications (Giannou et al., 2023; Fan et al., 2024; Barrault et al., 2024) aim to strengthen this capability by improving the expressivity of the transformer network and designing new supervision methods. Continuous CoT (Hao et al., 2024b; Zhu et al., 2025) replaces language-based CoT with continuous embeddings. It demonstrates that reasoning in latent space can break free from the constraint of discrete language tokens, allowing the model to encode superpositions of search paths in parallel and yielding efficiency gains. MVoT (Li et al., 2025b) trains a unified model to directly produce images and actions in interleaving trajectories. Recently, Mirage (Yang et al., 2025) proposes to empower VLM with latent visual tokens for multimodal reasoning, which uses the visual encoder of VLM for supervised fine-tuning using cosine similarity loss. In this work, we follow it and optimize the architecture by using diffusion decoder.

## 3 PRELIMINARY

**Vision Language Models.** VLMs extend a large language model (LLM) with visual perception so the model can "read" images as part of its context (Chen et al., 2024a). A standard VLM has three parts: (i) a vision encoder (*e.g.,* a ViT (Dosovitskiy et al., 2020)) that maps an input image to a sequence of visual features; (ii) a connector that projects these visual features into LLM-compatible "visual tokens", *i.e.,* continuous embeddings placed where tokens would appear; and (iii) the LLM backbone, which consumes a mixed sequence of visual and text tokens to produce a textual response. Given an image $I$ and text instruction $q$, the encoder yields features $\phi_v(I)$; the connector $g$ produces visual embeddings $\mathbf{z} = g(\phi_v(I))$; the LLM then attends over the concatenated sequence $[\mathbf{z}, q]$ and autoregressively generates an answer $y$. Training uses the standard next-token objective on a multimodal sequence $x = \{x_1, \ldots, x_n\}$, where each $x_i$ is a discrete text token:

$$P_\theta(x_{1:n}) = \prod_{i=1}^{n} P_\theta(x_i \mid x_{<i}, \mathbf{z}), \qquad \mathcal{L}(\theta) = -\sum_{i=1}^{n} \log P_\theta(x_i \mid x_{<i}, \mathbf{z}), \qquad (1)$$

so the model learns to predict each next token based on both linguistic context and visual evidence.

**Chain-of-thought.** CoT prompting is a widely used tactic for LLMs and VLMs to produce step-by-step solutions that make hard reasoning tasks easier. It works because the model is asked to externalize rationales (*i.e.,* short, structured intermediate steps), which organize relevant facts from the prompt, reduce search complexity, and keep the attention on task-critical context. In this way, CoT first generates a sequence of textual rationales and then an answer, denoted as $\{x_1, \cdots, x_n, a\}$.

In our setting, we extend CoT to modal-mixed CoT, where the reasoning trace interleaves natural-language tokens with compact latent visual embeddings. The model autoregressively generates the trace as $\{x_1, z_1, \cdots, z_n, x_n, a\}$. In this way, textual steps can invoke or reference latent visual states when helpful, enabling grounded, efficient multimodal reasoning.

## 4 APPROACH

We propose the modal-mixed chain-of-thought reasoning method that can interleave text tokens and visual latent embeddings, for solving vision-intensive complex tasks. To achieve it, we modify the

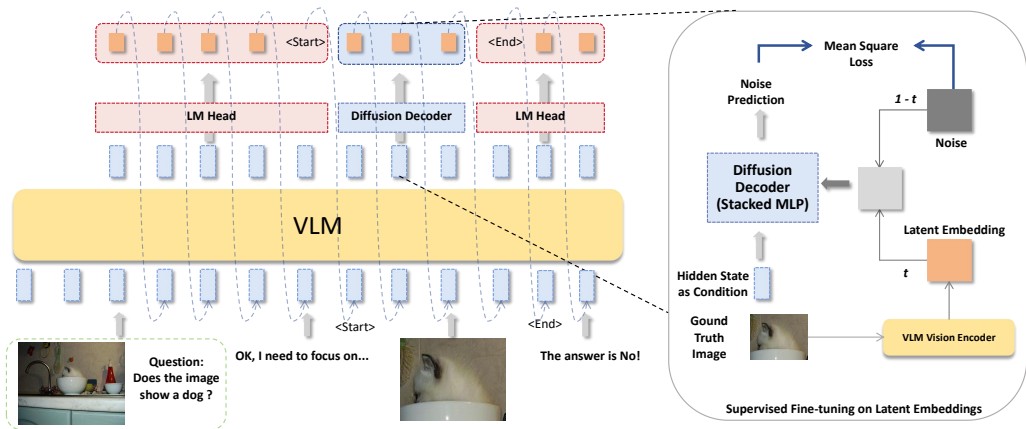

Figure 1: The overview of our proposed method. We integrate the diffusion model based decoder into the VLM and train it to learn the interleaved modal-mixed CoT reasoning paradigm.

original architecture of the VLM to support the new reasoning paradigm, and devise the fine-tuning strategy for learning it. The overview of the proposed method is shown in Figure 1.

## 4.1 MODEL ARCHITECTURE

Based on the original VLM architecture, we integrate it with a diffusion model based decoder (Yang et al., 2025) for generating latent visual embeddings, and add special tokens for performing modal-mixed chain-of-thought reasoning during inference.

**Diffusion Model based Decoder.** Inspired by diffusion models in vision generation (Rombach et al., 2022), we employ a conditional diffusion decoder to synthesize the latent visual embeddings that appear inside the modal-mixed CoT. Our design cleanly splits roles: the LLM produces high-level semantic intent, while a lightweight stacked-MLP diffusion network reconstructs fine-grained visual details conditioned on that intent. Concretely, the last-layer hidden state of the LLM serves as the conditioning vector for a token-wise diffusion process that transforms Gaussian noise into a latent embedding. Let $\mathbf{h}_k$ be the LLM's final hidden state when generating the $k$-th latent step; we map it to a conditioning vector $\mathbf{c}_k = \mathbf{W}\mathbf{h}_k$. Starting from $\mathbf{z}_k^{(T)} \sim \mathcal{N}(0, I)$, the decoder performs $T$ denoising steps using a small MLP predictor $\epsilon_\phi(\cdot)$ (with timestep and condition embeddings) to remove noise:

$$\mathbf{z}_k^{(t-1)} = \frac{1}{\sqrt{\alpha}} \left( \mathbf{z}_k^{(t)} - \frac{1 - \alpha_t}{\sqrt{1 - \bar{\alpha}_t}} \epsilon_\phi(\mathbf{z}_k^{(t)}, t, \mathbf{c}_k) \right) + \sigma_t \xi, \ \ \xi \sim \mathcal{N}(0, I), \tag{2}$$

and outputs the latent embedding $\mathbf{e}_k = \mathbf{z}_k^{(0)}$. To better adapt with the autoregressive language generation process, we emit these latents autoregressively. Specifically, after producing $\mathbf{e}_k$, we feed $\{\mathbf{e}_1, \ldots, \mathbf{e}_k\}$ back into the VLM to obtain the next condition $\mathbf{c}_{k+1}$, and repeat the diffusion loop to generate $\mathbf{e}_{k+1}$. Such a way unifies the generation paradigm of text tokens and latent embeddings in a next token/latent prediction manner, enabling joint optimization during training.

**Modal-mixed Chain-of-thought Reasoning.** Built on the diffusion decoder, our model performs modal-mixed CoT at inference by interleaving text tokens with latent visual embeddings. We introduce two special vocabulary items: ⟨START⟩ and ⟨END⟩, with randomly initialized embeddings. When generated, they switch the decoder between the standard language head and the diffusion latent head. Specifically, generation begins in the usual text mode; upon emitting ⟨START⟩, the model invokes the diffusion decoder to autoregressively produce a fixed number $K$ latent embeddings, then appends ⟨END⟩ and resumes text reasoning. A typical trajectory thus takes the form:

$$[\text{BOS}]; x_{1:i}; \langle\text{START}\rangle; \mathbf{z}_{1:K}; \langle\text{END}\rangle; x_{i+1:j} \cdots [\text{EOS}], \tag{3}$$

This scheme lets the model determine when to sketch, to help text and visuals mutually inform one another. During generation, text specifies what to sketch, and the latent visual embeddings refine and ground subsequent textual reasoning, to help text reasoning search the final answer.

## 4.2 Learning to Reason with Latent Embeddings

To enable the VLM to perform modal-mixed chain-of-thought reasoning, we employ a two-stage training strategy. First, we adopt supervised finetuning (SFT) to teach the model for learning the interleaved styled reasoning paradigm. Then, we apply reinforcement learning (RL) that can learn its generated modal-mixed CoTs, to further enhance its robustness and generalization capabilities.

**Supervised Fine-tuning with VLM as Visual Latent Encoder.** We first fine-tune the VLM on annotated, modal-interleaved CoT traces in which text rationales alternate with helpful intermediate images. For every intermediate image $\mathcal{I}$ in a trace, we reuse the VLM's own vision encoder and connection layer to convert it into dense visual token embeddings denoted as $\mathbf{V} = g(\phi_v(I)) \in \mathbb{R}^{N \times d}$. To reduce the context length while preserving high-level semantics, we compress $\mathbf{V}$ into a fixed-length latent sketch $\mathbf{z} \in \mathbb{R}^{M \times d}$ through the average pooling operation.

During training, the targeted sequence interleaves text tokens and compressed latent embeddings delimited by $\langle \text{START} \rangle$ and $\langle \text{END} \rangle$. We apply a joint autoregressive (AR) objective, where (i) the language head predicts the next text token; (ii) the diffusion decoder predicts the next latent embedding within the current latent block, conditioned on the VLM hidden state. For text positions $\mathcal{T}$ we minimize cross-entropy, and for each latent block $\mathbf{Z} = [\mathbf{z}_1, \ldots, \mathbf{z}_M]$, we sample $\boldsymbol{\epsilon} \sim \mathcal{N}(\mathbf{0}, \mathbf{I})$ and timestep $t$, form $\mathbf{z}_k^{(t)} = \alpha_t \mathbf{z}_k + \sigma_t \boldsymbol{\epsilon}$, and regress the diffusion decoder's velocity prediction $D_\phi(\mathbf{z}_k^{(t)}, t, \mathbf{c}_k)$ to $\boldsymbol{\epsilon}$. The joint loss is formulated as:

$$\mathcal{L} = -\sum_{t \in \mathcal{T}} \log p_\theta(y_t \,|\, x, y_{<t}, \mathbf{Z}_{\leq t}) \;+\; \lambda \sum_{k=1}^{M} \left\| D_\phi(\mathbf{z}_k^{(t)}, t, \mathbf{c}_k) - \boldsymbol{\epsilon} \right\|_2^2 \qquad (4)$$

where $\lambda$ balances textual next-token prediction and latent diffusion supervision.

**Reinforcement Learning for Self-adaptation.** Supervised fine-tuning (SFT) teaches the model to follow annotated, interleaved CoT traces. However, the provided intermediate images are not guaranteed to be the most helpful for textual reasoning, nor is the textual rationale necessarily ideal for guiding image (latent) generation. To better couple the two modalities, we add a reinforcement learning (RL) phase that lets the model roll out its own modal-mixed CoT and learn to prefer traces where text and latent sketches are mutually supportive. We adopt GRPO (Shao et al., 2024): for each query $q$, we sample a group of interleaved outputs $\{o_i\}_{i=1}^{G}$ from the current policy $\pi_{\theta_{old}}$, score each with a task accuracy reward $r_i$. For rewarding, we test the answer exactly matching accuracy, and assign 1 reward for correct ones and 0 for incorrect ones. Finally, we use the normalize advantages within the group, and optimize the clipped objective with a KL penalty to a reference policy:

$$\max_\theta \; \mathbb{E}\left[ \frac{1}{G} \sum_{i=1}^{G} \min\Big( \rho_i A_i, \; \text{clip}(\rho_i, 1 - \varepsilon, 1 + \varepsilon) A_i \Big) - \beta \, D_{\text{KL}}\big( \pi_\theta \,\|\, \pi_{\text{ref}} \big) \right], \qquad (5)$$

where $\rho_i = \frac{\pi_\theta(o_i|q)}{\pi_{\theta_{old}}(o_i|q)}$ and $A_i = \frac{r_i - \text{mean}(\{r_j\})}{\text{std}(\{r_j\})}$. We perform RL until convergence, which leads the VLM to discover modal-interleavings where latent visual reasoning genuinely aids the textual chain (and vice versa), yielding a self-adapted, tightly integrated modal-mixed CoT policy. To guarantee the training stability in RL, we do not backpropagate the loss from the latent visual embeddings, and only compute the loss on text tokens.

## 5 Experiments

### 5.1 Experimental Setup

**Evaluation Settings.** We evaluate our model from two perspectives: *vision-intensive perception* and *vision-intensive reasoning*. We select relevant perception and reasoning tasks from existing

| Model | VCog-Bench | | LogicVista | | MM-IQ | | | Average |
|---|---|---|---|---|---|---|---|---|
| | CVR | Raven | Ind. | Spat. | Math | Log. | 2D. | |
| **Qwen2.5-VL-7B-Instruct** | 39.8 | 12.1 | 25.2 | 19.0 | 24.5 | 24.4 | 25.2 | 24.3 |
| **LLaVA-OneVision-Qwen2-7B** | 36.7 | **13.0** | 20.6 | 25.3 | **29.1** | 23.5 | 25.0 | 24.7 |
| **InternVL2.5-8B** | 38.2 | 12.3 | 25.2 | 19.0 | 22.1 | 22.4 | 21.1 | 22.9 |
| **Janus-Pro-7B** | 34.5 | 12.5 | 20.4 | 27.9 | 26.5 | 19.6 | 22.7 | 23.4 |
| **Ours (SFT)** | 40.8 | 11.3 | **28.0** | **31.6** | 27.4 | 24.6 | 23.0 | **26.7** |
| **Ours (RL)** | **42.4** | 12.1 | 21.5 | 25.3 | 26.3 | **25.9** | **26.5** | 25.7 |

Table 1: Experimental Results on Vision-intensive Reasoning Task across VCog-Bench, LogicVista, and MM-IQ. Ind. Spat. Log. and 2D. denote inductive reasoning, spatial reasoning, logical operation, and 2D geometry, respectively. We also report the average result across all dimensions. The best and second best methods are marked by bold and underline, respectively.

benchmarks to compose the evaluation datasets, which typically require multi-round derivation on the vision input to reach the answer. For vision-intensive perception, we adopt the visual search benchmark $V^*$ (Wu & Xie, 2024), which is designed to rigorously assess fine-grained visual analysis under challenging conditions. Unlike conventional models that rely on external zoom-in functions, our model can directly perform intrinsic visual search. We further include perception subtasks from MME-Unify (Xie et al., 2025), a comprehensive benchmark for unified models. We focus on two subtasks: *Spot Difference* requiring localization of subtle differences between paired images, and *Auxiliary Lines* involving geometric guideline drawing and associated numerical reasoning. For vision-intensive reasoning, we employ three representative benchmarks. VCog-Bench (Cao et al., 2024) targets zero-shot abstract visual reasoning (AVR); we select two subsets: *CVR*, an outlier detection task over compositional visual patterns, and *RAVEN* (Zhang et al., 2019), an IQ-style matrix reasoning task for abstract rule induction and pattern completion. LogicVista (Xiao et al., 2024) evaluates visual logical reasoning by integrating perception with formal logic; we use the *Inductive* and *Spatial* subsets to assess abstract and visuospatial reasoning. Finally, MM-IQ (Cai et al., 2025) is inspired by human IQ tests and measures multimodal intelligence. We select the three largest and most representative subtasks—*Mathematical*, *2D Geometry*, and *Logical Operation*, which jointly cover numerical problem-solving, geometric reasoning, and symbolic logic, thereby providing a broad evaluation of multimodal reasoning capacity.

**Baseline Methods.** We compare our method against both traditional vision-language models and unified multimodal models. For traditional VLM baselines, we include three representative open-source VLMs: Qwen2.5-VL-7B-Instruct (Bai et al., 2025), LLaVA-OneVision-Qwen2-7B (Liu et al., 2024), and InternVL2.5-8B (Chen et al., 2024b). These VLMs follow the standard paradigm of coupling a vision encoder with a large language model via a connector, and are typically post-trained with visual instructions consisting of large-scale image captioning and visual questions, demonstrating state-of-the-art performance on most benchmarks. In addition, we evaluate Janus-Pro-7B (Chen et al., 2025b), a model that owns both image understanding and generation capabilities within a unified framework. Through a vision decoder, Janus-Pro is capable of generating full-resolution images and interleaves it with language reasoning. Our model can also be regarded as a unified model. One key difference is that, instead of generating images, our model produces compact visual imagery tokens that act as lightweight visual sketches to facilitate reasoning. Finally, we report results of our own models trained with supervised fine-tuning (SFT) and reinforcement learning (RL), namely Ours (SFT) and Ours (RL). Ours (SFT) reflects the performance gain learned from the instruction-following data alone, while the Ours (RL) further leverages preference optimization to improve reasoning robustness and output quality.

**Implementation Details.** In this work, we adopt Qwen2.5-VL-7B-Instruct as our base model and perform SFT using Zebra-CoT (Li et al., 2025a), a diverse large-scale dataset containing logically coherent interleaved text-image reasoning traces. Specifically, all images in Zebra-CoT are first resized to $448 \times 448$ and then passed through the vision encoder, producing 256 latent tokens per image, which are further compressed to 32 via average pooling. To enable the model to transition between text and latent reasoning, we introduce a loss term at the ⟨START⟩ token. Since latent

| Model | V* Benchmark | | MME-Unify | | Average |
| --- | --- | --- | --- | --- | --- |
| | Attr | Spatial | Spot Diff. | Aux. Lines | |
| **Qwen2.5-VL-7B-Instruct** | 72.2 | 77.6 | 13.0 | 32.7 | 48.8 |
| **LLaVA-OneVision-Qwen2-7B** | 60.0 | 67.1 | 27.0 | **46.2** | 50.0 |
| **InternVL2.5-8B** | 66.1 | 69.7 | **30.0** | 32.7 | 49.6 |
| **Janus-Pro-7B** | – | – | 29.0 | 28.8 | – |
| **Ours (SFT)** | 77.6 | **80.3** | 27.0 | 34.6 | 54.8 |
| **Ours (RL)** | **77.8** | 76.3 | 28.0 | 38.5 | **55.1** |

Table 2: Experimental Results on Vision-Intensive Perception Tasks. Spot Diff. and Aux. Lines are the abbreviation of the spot difference and auxiliary lines dimensions, respectively. Note that for VLMs in V* benchmark all use tools to help them achieve this good performance. The best and second-best methods are marked by bold and underline, respectively.

reasoning bypasses the language modeling head and cannot be directly detokenized, we insert a $\langle$PAD$\rangle$ placeholder whenever latent tokens are generated, in order to facilitate visualization. During training, we freeze the visual encoder and employ different learning rates for different components: the LLM uses a learning rate of 1e-5, while the diffusion head uses a learning rate of 2e-4, in order to achieve better convergence. We further adopt a cosine learning rate scheduler to stabilize optimization. For efficiency, instead of using the entire dataset, we select five representative subcategories, *i.e.,* Visual Jigsaw, Visual Search, Ciphers, RPM, and Tetris, resulting in 71,488 training samples. For RL, we leverage VisuLogic (Xu et al., 2025) as the training data and deduplicate it with the test data. It contains human-verified 1,000 high-quality visual reasoning problems across six categories. The sampling number is set to 500 per instance, and the learning rate is 5e-6.

## 5.2 MAIN RESULTS

Table 1 presents the results of vision-intensive reasoning tasks. As shown, Qwen2.5-VL-7B-Instruct and LLaVA-OneVision-Qwen2-7B perform better than Janus-Pro-7B on most of tasks. It indicates that VLMs typically own stronger reasoning capabilities than unified models. Although they can only generate language outputs, large-scale training empowers them better foundations for solving complicated tasks. Besides, our method (both SFT and RL versions) consistently outperforms or remains highly competitive with other baselines. For complex challenges such as the *CVR* in VCog-Bench or the *Inductive* and *Spatial* tasks in LogicVista, our model demonstrates significant improvements over the baselines. Traditional vision-language models rely on a single static encoding of the input image, which constrains their reasoning to a purely text-based chain of thought and prevents them from reinterpreting or redefining visual information, thereby limiting their effectiveness. Although Janus-Pro-7B is capable of generating both images and text, its lack of disentangling design also causes it to focus more on visual details rather than modal-mixed reasoning capability. By contrast, our model generates a sequence of intermediate latent embeddings interleaved with text tokens, and the disentanglement-based design using the lightweight diffusion decoder also liberates the VLM from visual details. Both guarantee the effectiveness of our method in accurate modal-mixed reasoning. Also, these latent visual rationales enable the decomposition of complex problems into manageable steps, leading to a much deeper and more effective level of reasoning.

Table 2 presents the results of vision-intensive perception tasks. For the V* benchmark, all the VLMs here use the zoom-in tools rely on external tools to obtain zoom-in views of local regions as auxiliary inputs, for helping achieve the good performance. Similar to reasoning tasks, Qwen2.5-VL-7B-Instruct exhibits better performance than other baselines, owing to its large-scale training on massive multimodal data. Janus-Pro-7B also demonstrates strong performance on the spot difference task of MME-Unify. Since this task requires to capture the visual details of the image, the unified model takes advantage from its learned visual details generation capability. As demonstrated, our method brings improvement to base model in many subtasks. For the tool-augmented VLMs, without such auxiliary images, their performance will drop significantly. In contrast, our model, equipped with latent reasoning capability, does not depend on external tools. By generating latent visual tokens, the model conducts visual search intrinsically rather than relying on auxiliary tools, resulting in stronger robustness and better overall performance.

| Model | V* Benchmark | | MME-Unify | | LogicVista | | Average |
|---|---|---|---|---|---|---|---|
| | Attr | Spatial | Spot Diff. | Aux. Lines | Ind. | Spat. | |
| **Ours (SFT)** | 77.6 | **80.3** | 27.0 | **34.6** | 28.0 | **31.6** | **46.5** |
| - Text-only Training | 77.6 | 71.1 | 21.0 | 26.9 | 21.5 | 25.3 | 40.6 |
| - Similarity Loss | **81.0** | 76.3 | **28.0** | 32.7 | **29.9** | 20.3 | 44.7 |
| Qwen2.5-VL-7B-Instruct | 72.2 | 77.6 | 13.0 | 32.7 | 25.2 | 19.0 | 40.0 |

Table 3: Ablation study results for text-only training and similarity loss.

| Model | MME-Unify | | LogicVista | | Average |
|---|---|---|---|---|---|
| | Spot Diff. | Aux. Lines | Ind. | Spat. | |
| **Qwen2.5-VL-7B-Instruct** | 13.0 | 32.7 | 25.2 | 19.0 | 22.5 |
| **Ours (SFT)** | **27.0** | **34.6** | **28.0** | **27.9** | **29.4** |
| **Language-only CoT** | 18.0 | 25.0 | 22.1 | 21.5 | 21.6 |

Table 4: Ability forgetting study results that test whether our fine-tuned VLM can perform the original language-only CoT reasoning.

By comparing the performance of our methods using SFT and RL, we can see that RL is more effective in improving the vision-intensive perception tasks than reasoning tasks. The reason is that the perception tasks typically require better fine-grained understanding of the image input, which needs the effective coordination between visual embedding and text token generation. However, RL leads to a significant degradation in the two dimensions from LogicVista benchmark. We observe that the two subtasks require to handle abstract logical patterns and the CoT output tends to be lengthy. A possible reason is that the long output pattern has not been well captured by the RL training data.

## 5.3 Further Analysis

**Ablation Study.** In our method, the latent visual embeddings within the modal-mixed CoT and the diffusion model based decoder are two key components. To study the effectiveness of these parts, we devise two variant models: (1) **Text-only Training** that only uses the text parts of the annotated CoT for training; (2) **Similarity Loss** that replaces the diffusion model based decoder by a MLP head, and directly uses the cosine similarity loss for optimizing the projected LLM output (Yang et al., 2025). We conduct the experiments on both the vision-intensive perception and reasoning datasets, *i.e.*, six subtasks from V* benchmark, MME-Unify and LogicVista. As shown in Table 3, our method consistently outperforms both the variant models and the base model on average. It indicates the effectiveness of both the latent embedding based modal-mixed CoT and the diffusion model based decoder. Among all the compared methods, the better results of the Similarity Loss model, together with our own method, demonstrate the effectiveness of latent–text interleaved reasoning. Moreover, the performance gap between our method and the Similarity Loss model highlights the importance of the diffusion decoder. By explicitly modeling the distribution of latent visual embeddings, the diffusion model based decoder enables the generation of higher-quality visual tokens, which serve as more informative and structured visual rationales, thereby enhancing reasoning capability.

**Catastrophic Forgetting Study.** In our approach, we use the visual encoder and connection layer from the VLM itself to supervise the training of our latent reasoning capability, and adopt a diffusion-based decoder for decomposing the high-level and low-level vision generation abilities. Such a well-aligned objective can avoid the catastrophic forgetting of the VLM for its original language-based reasoning ability. To study it, we remove the latent reasoning related modules (*i.e.,* diffusion model based decoder and special tokens), and use the VLM fine-tuned by our method for performing language-based CoT reasoning. The results in Table 4 shows that our model's language-based CoT reasoning capability has not been hurt a lot, with comparable performance with the backbone model. It demonstrates that our method can well keep the original knowledge and capability of the VLM, and also empowers it with a new modal-mixed CoT reasoning ability involving latent visual embeddings.

| Model | Token per Sample | Inference Latency (s) |
|---|---|---|
| **Qwen2.5-VL-7B-Instruct** | 219.30 | 18.31 |
| **Ours (SFT)** | 369.90 | 31.79 |

Table 5: Efficiency study that test the generated token number and inference latency per sample.

| $\lambda$ | V* Benchmark | | | LogicVista | | Average |
|---|---|---|---|---|---|---|
| | **Attr** | **Spatial** | **Overall** | **Ind.** | **Spat.** | |
| **0.1** | **85.3** | 77.6 | **82.3** | **30.1** | 15.2 | 58.1 |
| **1.0** | 77.6 | **80.3** | 78.6 | 28.0 | **31.6** | **59.2** |
| **10** | 75.9 | **80.3** | 75.9 | 25.7 | 20.3 | 55.6 |

Table 6: Hyperparameter tuning results of our method using different $\lambda$.

**Efficiency Study.** Although we add the latent embedding based vision reasoning process, it only needs the VLM to predict fixed-number tokens and passes a lightweight diffusion decoder. The inference latency is not increased a lot. To study it, we evaluate the computational efficiency of our proposed method against the Qwen2.5-VL-7B backbone using language-only CoT reasoning. Both models were evaluated on a single A100 GPU. The key performance, including token per sample and inference latency, are summarized in Table 5. We can observe that the increased cost is less than twice the original latency, which is acceptable in considering the increased 150 tokens per sample in average. In our model's latent reasoning process, each step that requires the generation of an intermediate visual embedding must invoke the multi-step denoising process in the diffusion decoder. Owing to its lightweight, it does not significantly increase the inference latency.

**Hyper-parameter Tuning.** A key component of our training objective is the hyperparameter $\lambda$ which balances textual next-token prediction and latent diffusion supervision. To understand the impact of this hyperparameter, we conducted an ablation study with $\lambda$ set to 0.1, 1.0, and 10. The results are presented in Table 6. A small $\lambda$ of 0.1 achieves the highest score on the attribute task V* benchmark, but for spatial task of LogicVista have low score. Increasing $\lambda$ to 1.0 slight decrease in performance on the attribute and inductive tasks, it dramatically boosts performance on complex spatial reasoning. The spatial task of LogicVista score increases to 31.6, and the spatial task V* benchmark score have a improvement to 80.3. However, increasing the weight further to $\lambda = 10$ proves to be detrimental. It significantly degrades performance across nearly all other metrics, we hypothesize that large $\lambda$ forces the model to focus too heavily on the visual latent embeddings, potentially makes reasoning performance drop. Based on this analysis, we selected $\lambda = 1.0$ for all our main experiments.

## 6 CONCLUSION

In this paper, we presented modal-mixed chain-of-thought (CoT), a new reasoning paradigm that enables a VLM to interleave language with compact visual "sketches" encoded as latent embeddings. To make these latents usable without eroding the original knowledge and capability of the VLM, we proposed to (1) train the VLM to reconstruct its own produced vision embeddings for intermediate images, and (2) attach a diffusion-based latent decoder that shoulders fine-grained perceptual detail while the language backbone supplies high-level intent. Based on the above architecture, we devised a two-stage training recipe, that first performs supervised fine-tuning with a joint next-token and latent objective then followed by reinforcement learning, to teaches the model what, when, and how to compose the long reasoning chains. Extensive experiments have demonstrated the effectiveness of our proposed method on 11 vision-intensive multimodal tasks.

In the future, we will extend the modal-mixed CoT strategy to additional modalities such as audio and 3D, for devising a unified modal-mixed CoT paradigm. Furthermore, we will develop uncertainty-aware policies that decide when to invoke visual reasoning and how intensively to sketch. In addition, we will continue to scale up the training data or add a pre-training stage on the new architecture, to explore the value of the new CoT paradigm for advancing machine intelligence beyond language.

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

## A  APPENDIX

You may include other additional sections here.

## USE OF LARGE LANGUAGE MODELS

In this work, large language models served a strictly supportive role. They were applied for minor editing tasks—improving readability, grammar, and flow—and for occasional debugging hints. Core contributions, including the conceptual framework, algorithm design, experimental setup, analysis, and conclusions, were fully developed and verified by the authors.

