# OpenReview forum: "Learning Modal-mixed Chain-of-thought Reasoning with Latent Embedding"
_ICLR.cc/2026/Conference — Submitted to ICLR 2026_

### Official Review · Reviewer_oDqR · 2025-10-25

**Soundness:** 2
**Presentation:** 2
**Contribution:** 3
**Rating:** 4
**Confidence:** 4

**Summary:**

This paper introduces a method to enable VLMs to interleave images (in latent space) into their CoT along with text. The main idea they used is to train a VLM to learn when to produce vision tokens, which are then converted to latent visual embeddings using a diffusion decoder. During training, they re-use the image encoder from the VLM to generate a fixed set of image embeddings which are trained using reconstruction loss after passing through the diffusion decoder along with the standard AR loss for text tokens. They also use GRPO to improve reasoning capabilities of the model. Their results show gains on visual reasoning and perception benchmarks over baselines and latency increases by approximately 70% due to the extra generation of visual embeddings.

**Strengths:**

1. The paper introduces a novel technique to allow VLMs to "think" in both text and visual spaces without the need for additional tools, allowing the model to visualize and render visual concepts freely.
2. The architecture and loss functions are intuitive and simple, and can be applied on top of any existing VLM with an image encoder by training it with the reconstruction loss.
3. The diffusion decoder and joint training ideas are ablated well and their methods show clear gains on a diverse set of multi-modal reasoning and perception benchmarks.

**Weaknesses:**

1. Lack of visualizations/qualitative depiction of visual latent embeddings: While their method shows improvements on several benchmarks, it is hard to understand what latent concepts the model is trying to visualize. It would be useful if the authors can either:
 a) run end-to-end diffusion from latents to pixels to see what images are actually rendered
 b) on some benchmark like V*, compare the cosine similarity (or any distance metric) between image outputs from tool-use (using the Qwen2.5VL baseline) and the image representations generated from their diffusion decoder.
 c) provide heat maps of similarity between patches on the image and the image embeddings to show which regions on the image are being focused on during the visual "thinking" process
2. Trade-off between latency and performance: There is a significant increase in the latency due to the image latent space generation process. It will be great if the authors can show a trade-off between performance and as it varies with
a) number of visual tokens rendered (K/M)
b) number of diffusion steps (T) (note: the value of T is also not mentioned in the paper
3. RL hurts performance: It is a little surprising that RL seems to decrease performance on benchmarks, especially on LogicVista, which should ideally improve with GRPO given that its a reasoning benchmark and longer CoT traces should generally help with better understanding of the problem, can the authors provide some additional RL training curves (reward, KL, solution length) to see why this might be happening
4. Missing clean comparison with tool-use: One of the claims of the paper is that latent space reasoning allows the model to "sketch" its own latents, but there is no clean quantitative comparison which clearly shows tool-use vs non tool-use baselines. The authors mention that in the V* benchmark the baselines have access to tools and their method doesn't but it will be good to have exact rows in the different benchmarks where the performance of existing baselines is measured with and without tool use.

**Questions:**

please check the weaknesses section above, in addition:

1. what is the value of T used during training and inference?
2. Are K (during inference) and M (during training kept the same)?
3. can you please include some sample completions (text tokens without image embeddings) from your method vs baselines on some queries to understand how the CoTs look like.
4. On average, how many times does the model invoke the <START> token to visualize something?

---

> ### Author Response · Authors · 2025-11-21
>
> Thank you for reviewing our paper. Our detailed responses to your comments are provided below.
>
> ## **Response to Weaknesses**
>
> W1:
>
> Thank you for the insightful suggestion. We agree that visualizations can substantially enhance the interpretability of our method. Since our latent tokens are highly compressed, directly computing similarity between image embeddings and raw image patches is not an appropriate comparison. Instead, we extract all attention maps corresponding to the 32 visual tokens at an intermediate layer (Layer 14), average them, and obtain an attention heatmap that reflects how the model attends to image tokens during visual “thinking.”
>
> Importantly, **we observe that the generated visual tokens consistently attend to the key regions required to answer the visual-search queries.** This provides direct evidence that the latent visual tokens indeed contribute meaningfully to the model’s reasoning process. The resulting heatmaps have been included in the supplementary PDF for review. Please have a check.
>
> W2:
>
> The comparison in Table 5 uses **Qwen2.5-VL-7B with language-only CoT** as the baseline. However, we realized this is not an ideal reference point: our model generates **more text tokens** after fine-tuning, so part of the latency difference is simply due to longer CoTs rather than the latent mechanism itself. To provide a fairer comparison, we conducted an additional analysis that isolates the true overhead of latent-token generation.Specifically, we compared three settings under controlled conditions:
>
> 10. **Text-only generation** of 32 tokens
>
> 11. **Latent-token generation** of 32 latent steps **(denoising steps set to 50\)**.
>
> 12. **Thyme**, a representative VLM that edits images by generating executable Python code. We measured from the moment the model outputs `<code>` token to the moment the edited image is re-fed into the VLM
>
> | Method | Seconds |
> | :---- | :---- |
> | Text-only generation (32) | 1.0311 |
> | Latent-token generation (32) | 3.1001 |
> | Thyme | 8.3575 |
>
> From the controlled comparison, latent-token generation takes roughly **0.10 s per latent**, which is slower than text-only generation but still **substantially faster than tool-use methods**. In particular, Thyme’s program-based pipeline—which requires code generation, external image editing, and re-encoding—incurs significantly higher latency. Thus, while latent CoT introduces moderate overhead relative to text CoT, it remains far more efficient than program-based visual reasoning.
>
> To further understand the cost–performance behavior of our method, we additionally evaluate how the number of latent tokens generated during inference affects both latency and performance on the LogicVista–Spatial subtask. This analysis isolates the effect of latent length from architecture or training differences and helps determine the optimal latent budget for spatial reasoning tasks.
>
> |  | seconds | spatial |
> | :---- | :---- | :---- |
> | Latent Tokens (64) | 6.0848 | 24.1 |
> | Latent Tokens (32) | 3.1001 | 31.6 |
> | Latent Tokens (16) | 1.5174 | 22.8 |
> | Latent Tokens (8) | 0.7886 | 19.0 |
> | Latent Tokens (4) | 0.3772 | 26.6 |
>
> We observe a clear trend: performance initially drops when very few latent tokens are generated, then improves as the latent sequence becomes longer, peaking at **3**, and decreases again when exceeding this length.
>
> This behavior is consistent with our training setup: the model was trained with **32 latent tokens**, so this length best matches the latent distribution it learned during SFT. Using fewer latents (e.g., 4–16) forces the model to compress complex spatial information too aggressively, leading to under-specified intermediate visual states and weaker reasoning. Conversely, generating more than 32 latents (e.g., 64\) introduces unnecessary noise and increases diffusion steps, producing diminishing returns and degraded performance.
>
> * **Zhang, Yi-Fan, et al. "Thyme: Think beyond images." *arXiv preprint arXiv:2508.11630* (2025).**
>
> W3:
>
> We agree that the empirical gains from the RL stage are not large. This is primarily due to current system-level bottlenecks rather than limitations of the method itself. In practice, existing inference frameworks such as vLLM do not yet support latent-mode generation, which makes each rollout extremely slow. As a result, we were only able to train on 500 samples and perform roughly 140 RL update steps in total—far fewer than typical RL settings. Addressing this limitation is part of our ongoing engineering work.

---

> > ### Comment · Reviewer_oDqR · 2025-11-22
> >
> > 1. Can you please provide some more implementation details on how the code execution and image editing is happening for Thyme? It seems surprising that code execution and image editing is taking more time than generating latents since they are run on the CPU whereas generating latents requires expensive forward passes.
> > 2. Can you please provide an ablation for the performance as it varies with number of latent tokens for other benchmarks as well?

---

> > > ### Author Response · Authors · 2025-11-22
> > >
> > > Q1:
> > >
> > > Thank you for the observation. You're correct \- the CPU operations are much faster than GPU forward passes. Our timing breakdown confirms this:
> > >
> > > * Code generation (GPU): \~8.3s (autoregressive generation of code tokens via forward passes)
> > > * Tool execution (CPU): \~0.02s (actual code execution/image editing)
> > > * Re-encoding (GPU): \~0.2s (feeding edited image back to VLM)
> > >
> > > The reported time is dominated by generating the code block through the model (\~8.3s), not executing it. The actual CPU-based image manipulation is negligible (\~20ms), which aligns with your intuition.
> > > We should clarify that "code execution and image editing time" includes the entire pipeline starting from code token generation. The bottleneck is the code block generation phase, not the CPU execution itself.
> > >
> > > Q2:
> > >
> > > We added another ablation experiments on the V\* benchmark. As shown in the table, we evaluate configurations ranging from 4 to 128 latent tokens on both the **Attr**  and **Spatial** subsets of V\*:
> > >
> > > | Model | V\* |  |
> > > | :---- | :---- | :---- |
> > > |  | **Attr**  | **Spatial** |
> > > | Latent Tokens (128) | 79.3 | 69.7 |
> > > | Latent Tokens (64) | 79.3 | **80.3** |
> > > | Latent Tokens (32) | 80.2 | 78.9 |
> > > | Latent Tokens (16) | 81.0 | 69.7 |
> > > | Latent Tokens (8) | **82.8** | 73.7 |
> > > | Latent Tokens (4) | 81.9 | 68.4 |
> > >
> > > **Key observations:**
> > >
> > > * The number of latent tokens has **minimal impact on the Attr subset**, with performance remaining stable around 79-82 across all configurations (4 to 128 tokens).
> > >
> > > * In contrast, the **Spatial subset is more sensitive** to the number of latent tokens, with performance ranging from 68.4 to 80.3. This indicates that spatial reasoning tasks require a more carefully tuned latent token budget.

---

> ### Author Response · Authors · 2025-11-22
>
> W4:
>
> On the V\* benchmark, all baseline methods **except ours** rely on the *zoom-in* tool; without this tool their performance drops substantially. In contrast, our method does **not** use any tool on V\*. For all other benchmarks in the paper, **no tool use is involved** for any model.
>
> Additionally, we have added **Thyme** as a new baseline—a model capable of editing images through generated Python code—to further strengthen the comparison.
>
> | Model | V\* |  | Logicvista |  | MMIQ |  |  |
> | :---- | :---- | :---- | :---- | :---- | :---- | :---- | :---- |
> |  | Attr  | Spatial | Ind.  | Spat.  | Math | Log.  | 2D. |
> | Qwen2.5-VL-7B-Instruct | 72.2 | 77.6 | 25.2 | 19.0 | 24.5 | 24.4 | 25.2 |
> | LLaVA-OneVision-Qwen2-7B | 60.0 | 67.1  | 20.6 | 25.3 | **29.1** | 23.5 | 25.0  |
> | InternVL2.5-8B | 66.1 | 69.7 | 25.2  | 19.0 | 22.1  | 22.4  | 21.1  |
> | Janus-Pro-7B | —— | —— | 20.4  | 27.9  | 26.5 | 19.6 | 22.7  |
> | R1-onevision-RL | 77.4 | **94.5** | 29.0 | 17.7 | 22.12 | 22.39 | 21.10 |
> | MM-EurekaQwen-7B | 41.7 | 64.5 | **34.6** | 22.8 | 25.11 | 23.66 | 24.93 |
> | Thyme | **82.5** | 78.9 | 33.6 | 19.0 | 24.1 | 20.6 | 26.3 |
> | Ours (SFT) | 77.6  | 80.3 | 28.0 | **31.6** | 27.4 | 24.6 | 23.0 |
> | Ours (RL) | 77.8 | 76.3 | 21.5  | 25.3 | 26.3 | **25.9**  | **26.5**  |
>
> ## **Analysis**
>
> Although all three newly added models are Qwen-based, their RL objectives differ drastically, resulting in expected trade-offs.
>  **R1-OneVision-RL** favors general logical reasoning but does not incorporate visual reasoning signals. **MM-Eureka-Qwen-7B** achieves strong math reasoning due to math-centric RL, but its perceptual and spatial abilities degrade correspondingly. Since both models rely purely on *text-only CoT*, they struggle on tasks that require integrating visual cues into the reasoning process. In contrast, our latent CoT naturally leverages spatial and perceptual information, leading to clear gains on tasks such as LogicVista–Spatial.
>
> Our comparison with **Thyme** highlights the fundamental difference between tool-use and latent-visual reasoning. Tool-use methods perform well when local, pixel-level edits (e.g., cropping, zooming) are sufficient, but they cannot represent hypothetical or future visual states because they operate strictly on the original pixels. Our model, however, generates **intermediate latent visual states**, allowing it to “imagine’’ spatial relations or geometric transformations that do not exist in the input image. This capability is crucial for complex spatial reasoning, as reflected in LogicVista–Spatial, where our model reaches **31.6** versus Thyme’s **19.0**.
>
> ## **Response to Question**
>
> Q1:
>
> During training, we use *T* \= 1000 diffusion steps. During inference, we adopt a shorter schedule with *T* \= 50 steps, which provides a good balance between performance and efficiency.
>
> Q2:
>
> Yes. In our implementation, we set M and  K to be the same. Keeping K \= M ensures that the diffusion decoder generates latent embeddings that match the shape and distribution seen during supervised training.
>
> Q3:
>
> We have included additional case studies in the Supplementary Material for your reference. Each case presents a comparison between our model's responses (labeled "Ours") and those from a standard text-based reasoning model (Qwen). Please have a check.
>
> Q4:
>
> The number of latent-token invocations varies by task type.
> For visual-search–style tasks such as **V\***—where a single localized inspection is typically sufficient—the model invokes the latent block **once** on average.
> For more complex reasoning benchmarks such as **LogicVista** or **MM-IQ**, the model tends to invoke latent tokens **2–4 times**, depending on the number of intermediate visual states required for multi-step reasoning.

---

### Official Review · Reviewer_QR9j · 2025-10-25

**Soundness:** 3
**Presentation:** 4
**Contribution:** 3
**Rating:** 6
**Confidence:** 4

**Summary:**

The paper presents a method to embed interleaved CoT to the latent space of a VLM. During training, the model is taught to reconstruct the generate the intermediate image that is relevant to the reasoning traces. During inference, the model generates visual embedding that is beneficial to the reasoning process. Experiment results on vision-centric reasoning benchmarks show the method improve SOTA VLMs like Qwen2.5-vl and InternVL-2.5.

**Strengths:**

* For text only input, methods like COCONUT use CoT data to iteratively embed the CoT into the latent space. This paper is conceptually interesting as it offers a promising approach to embed visual CoT to latent embeddings more explicitly and intuitively with a generative approach, as opposed to the previous cosine similarity approach.
* The paper is well written and clearly presents the motivation to extend the prior works on latent CoT by working with intermediate visual embedding instead of text CoT latents.

**Weaknesses:**

* Empirically, the RL stage yield only marginal gains, making it unclear what the purpose of this stage is.
* Based on the ablations, the latent CoT introduce noticeable amount of latency to inference. At the same time, the latent CoT also generates more tokens than the baseline. It would be important to conduct the experiment in a controlled compute budge to measure the latency vs performance relation of visual latent CoT and text CoT.
* Reasoning VLMs like R1-Onevision and MM-Eureka should be included as the baselines.

**Questions:**

* Could you clarify whether the text-only ablation contains interleave images from the zebra-cot data, only text, or both text and the original image? This is important because the fair comparison would be with models trained on the original zebra-cot data with interleaved image to demonstrate the effectiveness of the diffusion objective.
* It seems that the method is bottlenecked by the need of carefully curated interleaved CoT data. What are some potential solutions for such limitation?

---

> ### Author Response · Authors · 2025-11-21
>
> We appreciate your positive reviews. Our detailed responses to your comments are provided below.
>
> ## **Response to Weaknesses**
>
> W1:
>
> We agree that the empirical gains from the RL stage are not large. This is primarily due to current system-level bottlenecks rather than limitations of the method itself. In practice, existing inference frameworks such as vLLM do not yet support latent-mode generation, which makes each rollout extremely slow. As a result, we were only able to train on 500 samples and perform roughly 140 RL update steps in total—far fewer than typical RL settings. Addressing this limitation is part of our ongoing engineering work.
>
> W2:
>
>  The comparison in Table 5 uses **Qwen2.5-VL-7B with language-only CoT** as the baseline. However, we realized this is not an ideal reference point: our model generates **more text tokens** after fine-tuning, so part of the latency difference is simply due to longer CoTs rather than the latent mechanism itself. To provide a fairer comparison, we conducted an additional analysis that isolates the true overhead of latent-token generation.Specifically, we compared three settings under controlled conditions:
>
> 7. **Text-only generation** of 32 tokens
>
> 8. **Latent-token generation** of 32 latent steps **(denoising steps set to 50\)**.
>
> 9. **Thyme**, a representative VLM that edits images by generating executable Python code. We measured from the moment the model outputs `<code>` token to the moment the edited image is re-fed into the VLM
>
> | Method | Seconds |
> | :---- | :---- |
> | Text-only generation (32) | 1.0311 |
> | Latent-token generation (32) | 3.1001 |
> | Thyme | 8.3575 |
>
> From the controlled comparison, latent-token generation takes roughly **0.10 s per latent**, which is slower than text-only generation but still **substantially faster than tool-use methods**. In particular, Thyme’s program-based pipeline—which requires code generation, external image editing, and re-encoding—incurs significantly higher latency. Thus, while latent CoT introduces moderate overhead relative to text CoT, it remains far more efficient than program-based visual reasoning.
>
> To further understand the cost–performance behavior of our method, we additionally evaluate how the number of latent tokens generated during inference affects both latency and performance on the LogicVista–Spatial subtask. This analysis isolates the effect of latent length from architecture or training differences and helps determine the optimal latent budget for spatial reasoning tasks.
>
> |  | seconds | spatial |
> | :---- | :---- | :---- |
> | Latent Tokens (64) | 6.0848 | 24.1 |
> | Latent Tokens (32) | 3.1001 | 31.6 |
> | Latent Tokens (16) | 1.5174 | 22.8 |
> | Latent Tokens (8) | 0.7886 | 19.0 |
> | Latent Tokens (4) | 0.3772 | 26.6 |
>
> We observe a clear trend: performance initially drops when very few latent tokens are generated, then improves as the latent sequence becomes longer, peaking at **3**, and decreases again when exceeding this length.
>
> This behavior is consistent with our training setup: the model was trained with **32 latent tokens**, so this length best matches the latent distribution it learned during SFT. Using fewer latents (e.g., 4–16) forces the model to compress complex spatial information too aggressively, leading to under-specified intermediate visual states and weaker reasoning. Conversely, generating more than 32 latents (e.g., 64\) introduces unnecessary noise and increases diffusion steps, producing diminishing returns and degraded performance.
>
> * **Zhang, Yi-Fan, et al. "Thyme: Think beyond images." *arXiv preprint arXiv:2508.11630* (2025).**

---

> ### Author Response · Authors · 2025-11-22
>
> W3:
>
> We have added two **reasoning-oriented VLMs** trained with reinforcement learning: **R1-OneVision-RL** and **MM-Eureka-Qwen-7B**. Besides, we also added **Thyme**,  a representative tool-use VLM that edits images by generating executable Python code (cropping, drawing, zooming), serving as a clean baseline for explicit tool-invocation methods.
>
> | Model | V\* |  | Logicvista |  | MMIQ |  |  |
> | :---- | :---- | :---- | :---- | :---- | :---- | :---- | :---- |
> |  | Attr  | Spatial | Ind.  | Spat.  | Math | Log.  | 2D. |
> | Qwen2.5-VL-7B-Instruct | 72.2 | 77.6 | 25.2 | 19.0 | 24.5 | 24.4 | 25.2 |
> | LLaVA-OneVision-Qwen2-7B | 60.0 | 67.1  | 20.6 | 25.3 | **29.1** | 23.5 | 25.0  |
> | InternVL2.5-8B | 66.1 | 69.7 | 25.2  | 19.0 | 22.1  | 22.4  | 21.1  |
> | Janus-Pro-7B | —— | —— | 20.4  | 27.9  | 26.5 | 19.6 | 22.7  |
> | R1-onevision-RL | 77.4 | **94.5** | 29.0 | 17.7 | 22.12 | 22.39 | 21.10 |
> | MM-EurekaQwen-7B | 41.7 | 64.5 | **34.6** | 22.8 | 25.11 | 23.66 | 24.93 |
> | Thyme | **82.5** | 78.9 | 33.6 | 19.0 | 24.1 | 20.6 | 26.3 |
> | Ours (SFT) | 77.6  | 80.3 | 28.0 | **31.6** | 27.4 | 24.6 | 23.0 |
> | Ours (RL) | 77.8 | 76.3 | 21.5  | 25.3 | 26.3 | **25.9**  | **26.5**  |
>
> ## **Analysis**
>
> Although all three newly added models are Qwen-based, their RL objectives differ drastically, resulting in expected trade-offs.
>  **R1-OneVision-RL** favors general logical reasoning but does not incorporate visual reasoning signals. **MM-Eureka-Qwen-7B** achieves strong math reasoning due to math-centric RL, but its perceptual and spatial abilities degrade correspondingly. Since both models rely purely on *text-only CoT*, they struggle on tasks that require integrating visual cues into the reasoning process. In contrast, our latent CoT naturally leverages spatial and perceptual information, leading to clear gains on tasks such as LogicVista–Spatial.
>
> Our comparison with **Thyme** highlights the fundamental difference between tool-use and latent-visual reasoning. Tool-use methods perform well when local, pixel-level edits (e.g., cropping, zooming) are sufficient, but they cannot represent hypothetical or future visual states because they operate strictly on the original pixels. Our model, however, generates **intermediate latent visual states**, allowing it to “imagine’’ spatial relations or geometric transformations that do not exist in the input image. This capability is crucial for complex spatial reasoning, as reflected in LogicVista–Spatial, where our model reaches **31.6** versus Thyme’s **19.0**.
>
> * **Meng, Fanqing, et al. "Mm-eureka: Exploring the frontiers of multimodal reasoning with rule-based reinforcement learning." *arXiv preprint arXiv:2503.07365* (2025).**
> * **Yang, Yi, et al. "R1-onevision: Advancing generalized multimodal reasoning through cross-modal formalization." *arXiv preprint arXiv:2503.10615* (2025).**
> * **Zhang, Yi-Fan, et al. "Thyme: Think beyond images." *arXiv preprint arXiv:2508.11630* (2025).**
>
> ## **Response to Question**
>
> Q1:
>
> Our text-only ablation uses the text and original image but excludes the interleaved intermediate images, since Zebra-CoT does not provide text-only trajectories. To give a more appropriate comparison, and following your helpful suggestion we additionally include R1-OneVision and MM-EurekaQwen—both purely text-based reasoning models. The results show that adding latent visual embeddings can help in perception and spatial reasoning tasks, where textual reasoning alone is insufficient.
>
> Q2:
>
> Thank you for raising this concern. We recognize that curated interleaved CoT data is indeed a limiting factor for scaling. In this paper, we adopt a scalable approach by reusing automatically collected samples from Zebra-CoT, including embodied photographs and tool-generated images. This strategy enables us to collect larger, high-quality datasets without manual curation.
> Our experimental results demonstrate the effectiveness of this approach. As shown in the table below, continued training of the diffusion head with easily-collected images from the interleaved dataset yields consistent model improvements:
>
> | Model | V\* |  | Logicvista |  |
> | :---- | :---- | :---- | :---- | :---- |
> |  | Attr  | Spatial | Ind.  | Spat.  |
> | Ours | 77.6  | **80.3** | 28.0 | **31.6** |
> | Ours(continues training) | **83.6** | **80.3** | **31.9** | 29.6 |
>
> However, we acknowledge that Zebra-CoT's task diversity remains limited, and training diffusion models is inherently data-intensive. To address this limitation and enable the model to generate more dynamic visual aids, we are developing a tool-use-based agent system to generate high-quality, diverse data. Through this approach, we aim to train the model to generate auxiliary visual aids—such as zoom-in views, edge detection, and depth estimation—thereby enhancing its visual reasoning capabilities.

---

> > ### Author Response · Authors · 2025-11-27
> > **Visualization**
> >
> > To further validate the effectiveness of our diffusion loss, we extract all attention maps corresponding to the 32 visual tokens at an intermediate layer (Layer 14), average them, and obtain an attention heatmap that reflects how the model attends to image tokens during visual “thinking.” Importantly, **we observe that the generated visual tokens consistently attend to the key regions required to answer the visual-search queries**, providing direct evidence that the latent visual tokens meaningfully support the model’s reasoning process. **The resulting heatmaps have been included in the supplementary PDF; please take a look if you are interested.**

---

### Official Review · Reviewer_GxpB · 2025-10-31

**Soundness:** 2
**Presentation:** 3
**Contribution:** 2
**Rating:** 4
**Confidence:** 4

**Summary:**

This paper introduces modal-mixed chain-of-thought to enhance multimodal reasoning in VLMs. Unlike language-only COT, the approach interleaves textual tokens with latent visual embeddings. The architecture integrates a VLM with a diffusion-based decoder, aligning latent embeddings with the VLM's feature space to avoid modality gap issues. Experiments demonstrate the effectiveness of the approach.

**Strengths:**

1. Clear Motivation: The use of latent embeddings to form a vision-language interleaved chain of thought is a well-justified strategy. It effectively complements the strengths of pure-text chain-of-thought methods, presenting a reasonable approach for enhancing model performance in visual perception and reasoning tasks while maintaining inference efficiency.

2. Comprehensive Analysis and Ablations: The paper includes a thorough set of ablation studies. The analysis convincingly addresses key aspects such as catastrophic forgetting, computational efficiency, and hyperparameter tuning, providing strong empirical support for the proposed method.

3. High-Quality Writing and Presentation: The paper is well-written. The main figure is particularly intuitive and effectively illustrates the proposed framework.

**Weaknesses:**

1. Limited Novelty: For multimodal tasks, the chain-of-thought reasoning approach that involves either generating images or selecting image regions before answering questions has already been extensively explored in prior works [1][2][3]. In the embodied domain, methods that first generate latent or image representations before producing the final output have also been investigated  [4][5]. This limits the novelty of paper.

2. Insufficient Experiments:
  - Unfair Ablation Study: The paper indicates that the Zebra-COT dataset is used for training. However, examples from this dataset suggest that simply removing images results in linguistically incoherent reasoning, as subsequent reasoning steps may refer to previously referenced images. Proper post-processing of the text-only version of this dataset may be required to form a coherent reasoning chain. This affects the reliability of the ablation studies in Section 5.3.
  - Lack of Experiments on More Base Models: All experiments are conducted solely on the Qwen2.5-VL-Instruct model. This raises concerns about whether the method can generalize to other model families (e.g., InternVL, miniCPM) and models of varying parameter sizes.
  - Lack of Comparative Baselines: Apart from the JanusPro baseline, the other baseline models in the paper are evaluated directly without enhancements, making the baselines appear relatively weak. The authors should consider incorporating stronger or more advanced baselines (e.g., applying chain-of-thought prompting directly to these models, training models to autoregressively generate image tokens for reasoning, leveraging large models' attention mechanisms to insert image tokens into reasoning chains, or using soft tokens to train models in selecting visual regions before reasoning—all of which represent training-free vision-language chain-of-thought methods). At a minimum, for tasks solvable by directly selecting image tokens as part of the reasoning chain, the authors should strengthen comparisons and discussions with such approaches.

3. Lack of In-depth Discussion on Main Experimental Results: Table 1 shows that the proposed method does not lead to improvements across all benchmarks. Specifically, on the LogicVista and MM-IQ datasets, performance after RL training shows a noticeable decline compared to supervised fine-tuning (SFT). The paper lacks an analysis of the reasons behind these performance drops.

4. The authors may consider including more case studies to further strengthen the claims of the paper.

[1] Gao et al. (2025). "Interleaved-Modal Chain-of-Thought." arXiv preprint arXiv:2411.19488.

[2] Li et al. (2025). "Imagine while Reasoning in Space: Multimodal Visualization-of-Thought." arXiv preprint arXiv:2501.07542.

[3] Chen et al. (2025). "MINT-CoT: Enabling Interleaved Visual Tokens in Mathematical Chain-of-Thought Reasoning." arXiv preprint arXiv:2506.05331.

[4] Zhao et al. (2025). "CoT-VLA: Visual Chain-of-Thought Reasoning for Vision-Language-Action Models." arXiv preprint arXiv:2503.22020.

[5] Chen et al. (2025). "Moto: Latent Motion Token as the Bridging Language for Learning Robot Manipulation from Videos." arXiv preprint arXiv:2412.04445.

**Questions:**

Please see weaknesses.

---

> ### Author Response · Authors · 2025-11-21
>
> Thank you for reviewing our paper. Our detailed responses to your comments are provided below.
>
> ## **Response to Weaknesses**
>
> W1:
>
> We agree that both interleaved CoT and latent reasoning have been explored in prior work, but these approaches each have clear limitations that our method is designed to address.
>
> * **\[1\] and \[3\]** mainly perform local image operations, functioning as **visual-search** mechanisms rather than generating new visual content. In contrast, our self-generated latent sketches can go beyond local image and support **richer, model-internal visual transformations**. For example, we can predict future changes or world dynamics in an embodied setting—which is fundamentally beyond the local image operations. More Importantly, recent unified-model studies (e.g., Ming-UniVision) show that image generation and understanding mutually enhance each other. This provides empirical support for our design choice: equipping the model with its own generative pathway strengthens its visual representations, leading to better grounding and ultimately more effective visual reasoning.
>
> * **\[2\]** evaluate latent reasoning only within a single domain (e.g., VSP), making its methods **task-specialized rather than generalizable**. Our model explicitly targets broad generalization across diverse multimodal reasoning benchmarks.
>
> * **\[4\] and \[5\]** operate in robotic or embodied settings, where latents encode motion- or action-specific representations and thus have **much narrower visual semantics** than the general vision-reasoning tasks we study.
>
> Taken together, prior work addresses specific subproblems, while our contribution is a generalizable interleaved latent-reasoning framework capable of handling diverse and complex multimodal reasoning tasks.

---

> ### Author Response · Authors · 2025-11-27
>
> W2:
>
> For Unfair Ablation Study and Lack of Comparative Baselines
> We agree that removing images from Zebra-CoT inevitably produces partially incoherent text chains, as the dataset does not include a true text-only version. Our intention in this ablation was not to treat the text-only variant as a meaningful learning signal, but to show that our improvements cannot be explained simply by adding more textual reasoning. To further support this, we introduced two RL-trained reasoning VLMs(MM-EurekaQwen-7B and R1-onevision-RL). Their results on several benchmarks consistently demonstrate that text-only or text-focused reasoning alone cannot account for our gains, whereas latent visual embeddings provide clear additional benefits.
>
> We also add **Thyme**, a representative tool-use VLM that edits images by generating executable Python code (cropping, drawing, zooming), serving as a clean baseline for explicit tool-invocation methods.
>
> | Model | V\* |  | Logicvista |  | MMIQ |  |  |
> | :---- | :---- | :---- | :---- | :---- | :---- | :---- | :---- |
> |  | Attr  | Spatial | Ind.  | Spat.  | Math | Log.  | 2D. |
> | Qwen2.5-VL-7B-Instruct | 72.2 | 77.6 | 25.2 | 19.0 | 24.5 | 24.4 | 25.2 |
> | LLaVA-OneVision-Qwen2-7B | 60.0 | 67.1  | 20.6 | 25.3 | **29.1** | 23.5 | 25.0  |
> | InternVL2.5-8B | 66.1 | 69.7 | 25.2  | 19.0 | 22.1  | 22.4  | 21.1  |
> | Janus-Pro-7B | —— | —— | 20.4  | 27.9  | 26.5 | 19.6 | 22.7  |
> | R1-onevision-RL | 77.4 | **94.5** | 29.0 | 17.7 | 22.12 | 22.39 | 21.10 |
> | MM-EurekaQwen-7B | 41.7 | 64.5 | **34.6** | 22.8 | 25.11 | 23.66 | 24.93 |
> | Thyme | **82.5** | 78.9 | 33.6 | 19.0 | 24.1 | 20.6 | 26.3 |
> | Ours (SFT) | 77.6  | 80.3 | 28.0 | **31.6** | 27.4 | 24.6 | 23.0 |
> | Ours (RL) | 77.8 | 76.3 | 21.5  | 25.3 | 26.3 | **25.9**  | **26.5**  |
>
> ## **Analysis**
>
> Although all three newly added models are Qwen-based, their RL objectives differ drastically, resulting in expected trade-offs.
>  **R1-OneVision-RL** favors general logical reasoning but does not incorporate visual reasoning signals. **MM-Eureka-Qwen-7B** achieves strong math reasoning due to math-centric RL, but its perceptual and spatial abilities degrade correspondingly. Since both models rely purely on *text-only CoT*, they struggle on tasks that require integrating visual cues into the reasoning process. In contrast, our latent CoT naturally leverages spatial and perceptual information, leading to clear gains on tasks such as LogicVista–Spatial.
>
> Our comparison with **Thyme** highlights the fundamental difference between tool-use and latent-visual reasoning. Tool-use methods perform well when local, pixel-level edits (e.g., cropping, zooming) are sufficient, but they cannot represent hypothetical or future visual states because they operate strictly on the original pixels. Our model, however, generates **intermediate latent visual states**, allowing it to “imagine’’ spatial relations or geometric transformations that do not exist in the input image. This capability is crucial for complex spatial reasoning, as reflected in LogicVista–Spatial, where our model reaches **31.6** versus Thyme’s **19.0**.
>
> ## **Latency Comparison with Thyme**
>
>
> To provide a more comprehensive comparison, we conducted a controlled analysis to isolate the true computational overhead of latent-token generation. Specifically, we compared three settings under controlled conditions:
>
> 4. **Text-only generation** of 32 tokens
>
> 5. **Latent-token generation** of 32 latent steps **(denoising steps set to 50\)**.
>
> 6. **Thyme**, a representative VLM that edits images by generating executable Python code. We measured from the moment the model outputs `<code>` token to the moment the edited image is re-fed into the VLM
>
> | Method | Seconds |
> | :---- | :---- |
> | Text-only generation (32) | 1.0311 |
> | Latent-token generation (32) | 3.1001 |
> | Thyme | 8.3575 |
>
> From the controlled comparison, latent-token generation takes roughly **0.10 s per latent**, which is slower than text-only generation but still **substantially faster than tool-use methods**. In particular, Thyme’s program-based pipeline—which requires code generation, external image editing, and re-encoding—incurs significantly higher latency. Thus, while latent CoT introduces moderate overhead relative to text CoT, it remains far more efficient than program-based visual reasoning.
>
>
> * **Meng, Fanqing, et al. "Mm-eureka: Exploring the frontiers of multimodal reasoning with rule-based reinforcement learning." *arXiv preprint arXiv:2503.07365* (2025).**
> * **Yang, Yi, et al. "R1-onevision: Advancing generalized multimodal reasoning through cross-modal formalization." *arXiv preprint arXiv:2503.10615* (2025).**
> * **Zhang, Yi-Fan, et al. "Thyme: Think beyond images." *arXiv preprint arXiv:2508.11630* (2025).**

---

> ### Author Response · Authors · 2025-11-27
>
> Lack of Experiments on More Base Models
>
> We appreciate the reviewer's important point regarding generalization across model families. While adapting to InternVL and miniCPM would require substantial engineering effort due to code complexity, we have conducted additional experiments on **Qwen3-VL-4B-Instruct**. Notably, Qwen3-VL incorporates several structural improvements over Qwen2.5-VL, making it a meaningful validation of our method's generalizability. The results are as follows:
>
> | Model | V\* |  | Logicvista |  | MMIQ |  |  |
> | :---- | :---- | :---- | :---- | :---- | :---- | :---- | :---- |
> |  | Attr | Spatial | Ind. | Spat | Math | Log. | 2D. |
> | Qwen3-VL-4B-Instruct | 41.7 | 35.53 | 15.0 | 16.4 | **22.6** | 23.2 | 22.2 |
> | Ours | **76.7** | **77.6** | **27.4** | **25.3** | 21.6 | **25.1** | **29.8** |
>
> These results demonstrate substantial improvements over the baseline model, validating the effectiveness of our method across models of different parameter scales. This provides evidence that our approach can generalize beyond the specific model used in the main paper.
>
> W3:
>
> We agree that the empirical gains from the RL stage are not large. This is primarily due to current system-level bottlenecks rather than limitations of the method itself. In practice, existing inference frameworks such as vLLM do not yet support latent-mode generation, which makes each rollout extremely slow. As a result, we were only able to train on 500 samples and perform roughly 140 RL update steps in total—far fewer than typical RL settings. Addressing this limitation is part of our ongoing engineering work.
>
> W4:
>
> We have included additional case studies in the Supplementary Material for your reference. Each case presents a comparison between our model's responses (labeled "Ours") and those from a standard text-based reasoning model (Qwen). Please have a check.
>
> Besides, to further validate the effectiveness of our diffusion loss, we extract all attention maps corresponding to the 32 visual tokens at an intermediate layer (Layer 14), average them, and obtain an attention heatmap that reflects how the model attends to image tokens during visual “thinking.” Importantly,**we observe that the generated visual tokens consistently attend to the key regions required to answer the visual-search queries, providing direct evidence that the latent visual tokens meaningfully support the model’s reasoning process.** The resulting heatmaps have been included in the supplementary PDF; please take a look if you are interested.

---

### Official Review · Reviewer_tYa2 · 2025-11-05

**Soundness:** 2
**Presentation:** 3
**Contribution:** 2
**Rating:** 2
**Confidence:** 4

**Summary:**

This paper introduces a modal-mixed chain-of-thought reasoning approach that enables VLMs to interleave textual tokens with compact visual "sketches" represented as latent embeddings. The method addresses the limitation that language-only CoT struggles on vision-intensive problems where intermediate states are inherently visual. The approach consists of two key architectural components: (1) using the VLM's own vision encoder to produce latent embeddings for intermediate visual states, ensuring semantic alignment, and (2) a diffusion-based decoder that generates these latent embeddings conditioned on the VLM's hidden states. Training proceeds in two stages: supervised fine-tuning on modal-interleaved traces with joint next-token and latent-reconstruction objectives, followed by reinforcement learning to learn when to switch modalities. Evaluated on 11 vision-intensive tasks spanning perception and reasoning, the method demonstrates improvements over language-only CoT and competitive baselines.

**Strengths:**

1. The motivation is compelling and well-articulated. The observation that humans use mental imagery and visuospatial reasoning for vision-intensive tasks provides strong intuition for why purely language-based CoT may be insufficient. The analogy to a "visuospatial sketchpad" is effective and the problem is clearly motivated.

2. The architectural design is clear and well-motivated. Using the VLM's own vision encoder to produce target latent embeddings is a clever way to learn visual representations for the task.

3. The two-stage training strategy is well-designed. The SFT stage teaches the basic modal-interleaved format while the RL stage allows the model to discover effective switching points between modalities. The joint objective combining next-token prediction with latent reconstruction during SFT is natural and principled.

4. The paper is generally well-written and clearly structured, making the method easy to follow.

**Weaknesses:**

1. The most critical weakness is the complete absence of comparisons with existing visual tool-use and visual generation-based reasoning methods, which is a major gap given that this is the paper's core contribution area. There is extensive recent work on using visual information as intermediate reasoning steps, including: Visual Sketchpad (Hu et al., 2024), CoT-VLA (Zhao et al., 2025), Thinking with Generated Images (Chern et. al.), and Imagination and Contemplation (Christoduolopoulos et. al.) The broader landscape of visual tool-use methods documented in the recent survey "Thinking with Images for Multimodal Reasoning" (arxiv 2506.23918), which categorizes tool-augmented visual reasoning, visual programming (ViperGPT, VisProg), and other paradigms. Without comparisons to these similar approaches, readers cannot assess whether latent embeddings provide advantages over: (i) explicit visual tool use with actual image manipulation, (ii) code-based visual generation, or (iii) other forms of visual intermediate reasoning where an image generation model is used as an accessible LLM agent. The paper claims latent embeddings are better because they're "lightweight" but provides no empirical evidence for these claims against actual tool-use baselines.

2. Related to the above, the paper's positioning relative to existing work is incomplete. The related work section (Section 2) mentions "Recent works have proposed to use interleaved CoT for solving vision-intensive tasks (Gao et al., 2025)" and cites Wang et al. 2025a for tool use, but then dismisses these with "such a way cannot handle open questions that might require special operations on the images (e.g., drawing irregular masks)." This dismissal is unfounded—Visual Sketchpad and similar methods can handle diverse visual operations including masks, bounding boxes, and arbitrary drawings. The paper also mentions Mirage (Yang et al., 2025) only briefly in the latent reasoning paragraph, despite it being the most similar prior work. The relationship needs thorough clarification and at least some empirical comparisons to a subset of related works.

3. The improvements over baselines are modest and inconsistent across tasks. Table 1 shows the method achieves 26.7 (SFT) and 25.7 (RL) average scores compared to 24.3-24.7 for baselines: gains of around 2 points. Moreover, RL actually hurts performance on several tasks (Inductive reasoning drops from 28.0 to 21.5, Spatial reasoning from 31.6 to 25.3). The paper attributes this to "lengthy output patterns not well captured by RL data" but this suggests brittleness. Given these modest and inconsistent gains, stronger baselines using visual reasoning methods are essential to validate the approach.

4. The computational costs are substantial but inadequately analyzed. Table 5 shows inference latency increases from 18.31s to 31.79s (73% increase) and tokens per sample from 219 to 370. The paper dismisses this as "acceptable" without cost-benefit analysis. How does this compare to tool-use methods? To other multimodal CoT methods?

5. On this note, one dimension of comparison left out are multimodal CoT methods (such as Compositional CoT by Mitra et. al. and many others that can be found https://github.com/BradyFU/Awesome-Multimodal-Large-Language-Models) and latent CoT approaches like COCONUT (Hao et. al.). Though less comparable, it would be great to see a comparison to these CoT methods.

**Questions:**

My feedback is constructive and includes both critique and approaches for resolving that critique. Please refer to the Weaknesses section.

---

> ### Author Response · Authors · 2025-11-21
>
> Thank you for reviewing our paper. Our detailed responses to your comments are provided below.
>
> ## **Response to Weaknesses**
>
> W1:
>
> To address your concern about insufficient baselines, we include three additional models in our updated evaluation.
>
> We add two **reasoning-oriented VLMs** trained with reinforcement learning: **R1-OneVision-RL** (optimized with broad cross-domain RL) and **MM-Eureka-Qwen-7B** (trained almost entirely on math-focused RL signals) and **Thyme**, a representative tool-use VLM that edits images by generating executable Python code (cropping, drawing, zooming), serving as a clean baseline for explicit tool-invocation methods.
>
> | Model | V\* |  | Logicvista |  | MMIQ |  |  |
> | :---- | :---- | :---- | :---- | :---- | :---- | :---- | :---- |
> |  | Attr  | Spatial | Ind.  | Spat.  | Math | Log.  | 2D. |
> | Qwen2.5-VL-7B-Instruct | 72.2 | 77.6 | 25.2 | 19.0 | 24.5 | 24.4 | 25.2 |
> | LLaVA-OneVision-Qwen2-7B | 60.0 | 67.1  | 20.6 | 25.3 | **29.1** | 23.5 | 25.0  |
> | InternVL2.5-8B | 66.1 | 69.7 | 25.2  | 19.0 | 22.1  | 22.4  | 21.1  |
> | Janus-Pro-7B | —— | —— | 20.4  | 27.9  | 26.5 | 19.6 | 22.7  |
> | R1-onevision-RL | 77.4 | **94.5** | 29.0 | 17.7 | 22.12 | 22.39 | 21.10 |
> | MM-EurekaQwen-7B | 41.7 | 64.5 | **34.6** | 22.8 | 25.11 | 23.66 | 24.93 |
> | Thyme | **82.5** | 78.9 | 33.6 | 19.0 | 24.1 | 20.6 | 26.3 |
> | Ours (SFT) | 77.6  | 80.3 | 28.0 | **31.6** | 27.4 | 24.6 | 23.0 |
> | Ours (RL) | 77.8 | 76.3 | 21.5  | 25.3 | 26.3 | **25.9**  | **26.5**  |
>
> ## **Analysis**
>
> Although all three newly added models are Qwen-based, their RL objectives differ drastically, resulting in expected trade-offs.
>  **R1-OneVision-RL** favors general logical reasoning but does not incorporate visual reasoning signals. **MM-Eureka-Qwen-7B** achieves strong math reasoning due to math-centric RL, but its perceptual and spatial abilities degrade correspondingly. Since both models rely purely on *text-only CoT*, they struggle on tasks that require integrating visual cues into the reasoning process. In contrast, our latent CoT naturally leverages spatial and perceptual information, leading to clear gains on tasks such as LogicVista–Spatial.
>
> Our comparison with **Thyme** highlights the fundamental difference between tool-use and latent-visual reasoning. Tool-use methods perform well when local, pixel-level edits (e.g., cropping, zooming) are sufficient, but they cannot represent hypothetical or future visual states because they operate strictly on the original pixels. Our model, however, generates **intermediate latent visual states**, allowing it to “imagine’’ spatial relations or geometric transformations that do not exist in the input image. This capability is crucial for complex spatial reasoning, as reflected in LogicVista–Spatial, where our model reaches **31.6** versus Thyme’s **19.0**.
>
> * **Meng, Fanqing, et al. "Mm-eureka: Exploring the frontiers of multimodal reasoning with rule-based reinforcement learning." *arXiv preprint arXiv:2503.07365* (2025).**
> * **Yang, Yi, et al. "R1-onevision: Advancing generalized multimodal reasoning through cross-modal formalization." *arXiv preprint arXiv:2503.10615* (2025).**
> * **Zhang, Yi-Fan, et al. "Thyme: Think beyond images." *arXiv preprint arXiv:2508.11630* (2025).**
>
> W2:
> We thank the reviewer for highlighting the need to clarify our positioning.
>
> First, we believe self-generated visual aids offer substantially stronger generalization than tool-based pipelines. They allow the model to internally simulate arbitrary visual states—for example, predicting future changes or world dynamics in an embodied setting—which is fundamentally beyond the capability of a fixed set of external visual tools. Even when invoking powerful tools such as video-generation models, external calls remain less stable and introduce significantly higher latency compared to native latent generation integrated directly into the VLM. More Importantly, recent unified-model studies (e.g., **Ming-UniVision: Joint Image Understanding and Generation with a Unified Continuous Tokenizer**) show that **image generation and image understanding mutually enhance each other**. This provides empirical support for our design choice: equipping the model with its own generative pathway strengthens its visual representations, leading to better grounding and ultimately more effective visual reasoning.
>
> Second, while our work is inspired by Mirage (Yang et al., 2025), Mirage evaluates latent visual reasoning mainly on VSP and Jigsaw tasks with training and test data drawn from the same domain, limiting its generalizability. Our goal is to develop **generalizable interleaved latent reasoning** that works across diverse benchmarks. In addition, our ablations demonstrate that our diffusion-based latent decoder is more effective than the cosine-similarity loss introduced in Mirage, confirming the benefit of our architectural design.

---

> > ### Author Response · Authors · 2025-11-22
> >
> > W3:
> > We agree that the empirical gains from the RL stage are not large. This is primarily due to current system-level bottlenecks rather than limitations of the method itself. In practice, existing inference frameworks such as vLLM do not yet support latent-mode generation, which makes each rollout extremely slow. As a result, we were only able to train on 500 samples and perform roughly 140 RL update steps in total—far fewer than typical RL settings. Addressing this limitation is part of our ongoing engineering work.
> >
> > W4:
> >  The comparison in Table 5 uses **Qwen2.5-VL-7B with language-only CoT** as the baseline. However, we realized this is not an ideal reference point: our model generates **more text tokens** after fine-tuning, so part of the latency difference is simply due to longer CoTs rather than the latent mechanism itself. To provide a fairer comparison, we conducted an additional analysis that isolates the true overhead of latent-token generation.Specifically, we compared three settings under controlled conditions:
> >
> > 4. **Text-only generation** of 32 tokens
> >
> > 5. **Latent-token generation** of 32 latent steps **(denoising steps set to 50\)**.
> >
> > 6. **Thyme**, a representative VLM that edits images by generating executable Python code. We measured from the moment the model outputs `<code>` token to the moment the edited image is re-fed into the VLM
> >
> > | Method | Seconds |
> > | :---- | :---- |
> > | Text-only generation (32) | 1.0311 |
> > | Latent-token generation (32) | 3.1001 |
> > | Thyme | 8.3575 |
> >
> > From the controlled comparison, latent-token generation takes roughly **0.10 s per latent**, which is slower than text-only generation but still **substantially faster than tool-use methods**. In particular, Thyme’s program-based pipeline—which requires code generation, external image editing, and re-encoding—incurs significantly higher latency. Thus, while latent CoT introduces moderate overhead relative to text CoT, it remains far more efficient than program-based visual reasoning.
> >
> > To further understand the cost–performance behavior of our method, we additionally evaluate how the number of latent tokens generated during inference affects both latency and performance on the LogicVista–Spatial subtask. This analysis isolates the effect of latent length from architecture or training differences and helps determine the optimal latent budget for spatial reasoning tasks.
> >
> > |  | seconds | spatial |
> > | :---- | :---- | :---- |
> > | Latent Tokens (64) | 6.0848 | 24.1 |
> > | Latent Tokens (32) | 3.1001 | 31.6 |
> > | Latent Tokens (16) | 1.5174 | 22.8 |
> > | Latent Tokens (8) | 0.7886 | 19.0 |
> > | Latent Tokens (4) | 0.3772 | 26.6 |
> >
> > We observe a clear trend: performance initially drops when very few latent tokens are generated, then improves as the latent sequence becomes longer, peaking at **3**, and decreases again when exceeding this length.
> >
> > This behavior is consistent with our training setup: the model was trained with **32 latent tokens**, so this length best matches the latent distribution it learned during SFT. Using fewer latents (e.g., 4–16) forces the model to compress complex spatial information too aggressively, leading to under-specified intermediate visual states and weaker reasoning. Conversely, generating more than 32 latents (e.g., 64\) introduces unnecessary noise and increases diffusion steps, producing diminishing returns and degraded performance.
> >
> > * **Zhang, Yi-Fan, et al. "Thyme: Think beyond images." *arXiv preprint arXiv:2508.11630* (2025).**

---

> > > ### Comment · Reviewer_tYa2 · 2025-11-26
> > >
> > > I thank the authors for taking the time to respond to my feedback. In light of the added baselines and more thorough explanation of the positioning, I am willing to raise my score. If accepted, please do incorporate the new results, explanation, and discussion of latency and efficiency in the camera-ready paper.

---

> ### Author Response · Authors · 2025-11-26
>
> We sincerely thank the reviewer for carefully revisiting our submission and for raising the score. The newly added baselines and experiments have indeed strengthened the work. The reviewer’s suggestions motivated us to conduct more comprehensive evaluations, through which we uncovered important insights into the latency–efficiency trade-offs. We will incorporate all new results and discussions into the camera-ready version, should the paper be accepted. Thank you again for the thoughtful and constructive feedback.

---

### Meta-Review · Area_Chair_3evw · 2026-01-03

**Summary:**

This paper studies how to extend chain-of-thought (CoT) reasoning beyond pure text for multimodal reasoning tasks. While textual CoT is effective for many problems, it often struggles when key intermediate reasoning steps are visual in nature. The authors propose a modal-mixed CoT framework that interleaves language with compact latent visual representations, allowing the model to reason using both modalities. The approach reuses the vision–language model as its own visual encoder to maintain semantic alignment, and introduces a diffusion-based latent decoder to generate visual embeddings while keeping high-level reasoning in the language backbone. However, some reviewers pointed out that the novelty of this work is limited, as related ideas have already been explored in prior studies. In addition, the experimental evaluation is conducted on only a small set of base models, which raises concerns about the generalizability of the conclusions.

**Reviewer Concerns:**

Reviewer tYa2 indicated that they would raise their score.

**Reviewer Scores:**

Reviewer oDqR may raise their score after the discussion, as the authors addressed all of their questions.

---

### Decision · Program_Chairs · 2026-01-26

Reject